# Molecular Surveillance for Bocaparvoviruses and Bufaviruses in the European Hedgehog (*Erinaceus europaeus*)

**DOI:** 10.3390/microorganisms12010189

**Published:** 2024-01-17

**Authors:** Vittorio Sarchese, Andrea Palombieri, Ilaria Prandi, Serena Robetto, Luigi Bertolotti, Maria Teresa Capucchio, Riccardo Orusa, Mitzy Mauthe von Degerfeld, Giuseppe Quaranta, Massimo Vacchetta, Vito Martella, Barbara Di Martino, Federica Di Profio

**Affiliations:** 1Department of Veterinary Medicine, Università degli Studi di Teramo, 64100 Teramo, TE, Italy; vsarchese@unite.it (V.S.); apalombieri@unite.it (A.P.); bdimartino@unite.it (B.D.M.); 2Centro Animali Non Convenzionali (C.A.N.C.), Department of Veterinary Sciences, University of Turin, 10095 Grugliasco, TO, Italy; ilaria.prandi@unito.it (I.P.); mariateresa.capucchio@unito.it (M.T.C.); mitzy.mauthe@unito.it (M.M.v.D.); giuseppe.quaranta@unito.it (G.Q.); 3Centro di Referenza Nazionale per le Malattie degli Animali Selvatici (CeRMAS), Istituto Zooprofilattico Sperimentale del Piemonte, della Liguria e della Valle d’Aosta, 11020 Quart, AO, Italy; serena.robetto@izsto.it (S.R.); riccardo.orusa@izsto.it (R.O.); 4Department of Veterinary Sciences, University of Turin, 10095 Grugliasco, TO, Italy; luigi.bertolotti@unito.it; 5Centro Recupero Ricci “La Ninna”, 12060 Novello, CN, Italy; maxvacchetta@live.it; 6Department of Veterinary Medicine, Università Aldo Moro di Bari, 70010 Valenzano, BA, Italy; vito.martella@uniba.it

**Keywords:** bocaparvoviruses, bufaviruses, hedgehogs, duodenum, liver

## Abstract

The presence of bocaparvoviruses (BoVs) and bufaviruses (BuVs) in the European hedgehog (*Erinaceus europaeus*) was investigated by screening duodenal and liver samples collected from 183 carcasses, delivered to wildlife rescue centers located in northwestern Italy. BoV DNA was detected in 15 animals (8.2%), with prevalences of 7.1% (13/183) and 2.7% (5/183) in intestine and liver samples, respectively. Upon the sequence analyses of the NS1 gene, two highly divergent BoVs (65.5–67.8% nt identities) were identified. Fourteen strains showed the highest identity (98.3–99.4% nt) to the hedgehog BoV strains recently detected in China in Amur hedgehogs (*Erinaceus amurensis*), whilst four strains were genetically related (98.9–99.4% nt identities) to the porcine BoVs identified in pigs and classified in the species *Bocaparvovirus ungulate 4*, which included related viruses also found in rats, minks, shrews, and mice. BuV DNA was detected in the duodenal samples of two hedgehogs, with a prevalence rate of 1.1%. The nearly full-length genome of two BuV strains, Hedgehog/331DU-2022/ITA and Hedgehog/1278DU/2019/ITA, was reconstructed. Upon phylogenetic analysis based on the NS and VP aa sequences, the Italian hedgehog BuVs tightly clustered with the BuVs recently identified in the Chinese Amur hedgehogs, within a potential novel candidate species of the genus *Protoparvovirus*.

## 1. Introduction

The *Parvoviridae* family includes a remarkably diverse group of viruses, classified into three subfamilies. *Parvovirinae* and *Densovirinae* contain viruses that infect vertebrates and invertebrates, respectively, whilst the subfamily *Hamaparvovirinae* includes viruses identified in both vertebrate and invertebrate hosts [1,2]. Parvoviruses are small, non-enveloped viruses with an icosahedral capsid surrounding a positive-sense single-stranded (ss) DNA genome of 3.9–6.3 kb. The coding region, flanked at the 5′ and 3′ ends by two terminal repeats that fold into complex hairpin-like structures, is organized in two major expression cassettes with open reading frames (ORFs) encoding for non-structural (NS) proteins on the left-hand side and structural viral proteins (VPs) on the right-hand side [3].

In recent years, thanks to improvements in viral discovery approaches, the number of newly identified parvoviruses in diseased and healthy animals has expanded rapidly as the number of known host species [4]. By using broad-range consensus primers and high-throughput sequencing, novel members of the genera *Bocaparvovirus* and *Protoparvovirus* have been identified in dogs [5,6,7]; cats [8,9,10,11]; and wildlife, including pine martens [12], minks [13], sea otters [14], wolves [15], and foxes [15,16,17]. More recently, during a large-scale survey aimed to investigate the diversity and abundance of viruses in game animals in China, novel parvoviruses, named hedgehog bocaviruses (HhBoVs) and hedgehog bufaviruses (HhBuVs), genetically related to members of the genera *Bocaparvovirus* and *Protoparvovirus (Protoparvovirinae* subfamily), respectively, and to a hedgehog chaphamaparvovirus (genus *Chaphamaparvovirus* of the *Hamaparvovirinae* subfamily), have been identified in the Amur hedgehog *Erinaceus amurensis* Schrenk, 1858, using a metagenomic approach [18]. However, the epidemiological patterns of these novel viruses were not characterized. Whether these viruses are common in hedgehogs or they were detected serendipitously is unknown. Interestingly, chaphamaparvoviruses genetically related to the strain detected in the Chinese study [18] have been identified through a metaviromic investigation in the European hedgehog *Erinaceus europaeus* Linnaeus, 1758, during an outbreak of enteritis in an Italian wildlife rescue center (WRC) [19]. The presence of parvoviruses in *E. europaeus* has been already documented. During an outbreak of acute gastroenteritis by the feline parvovirus (FPV) (genus *Protoparvovirus*, species *Protoparvovirus Carnivoran 1*) in cats housed with European hedgehogs, reactivity via immunohistochemistry was detected using an FPV polyclonal antibody in the intestine of the hedgehogs died of enteritis [20].

Overall, based on the limited available literature, European hedgehogs might host genetically diverse parvoviruses. Herein, we performed a molecular investigation for BoVs and BuVs in European hedgehogs admitted to two WRCs in northwestern Italy (Piedmont Region). The nearly full-length genome of two HhBuV strains was determined.

## 2. Materials and Methods

### 2.1. Sampling

The sample included a total of 183 carcasses of *E. europaeus* subjected to complete necropsy during a time frame spanning from 2018 to 2022. Of these, 146 hedgehogs were admitted at the “Centro Animali Non Convenzionali (C.A.N.C.)” of the Department of Veterinary Sciences, University of Turin, between 2018 and 2022, whereas 37 additional animals recovered in 2022 at the “La Ninna” WRC, a center specialized in the treatment and rehabilitation of European hedgehogs. All animals were found sick, injured, or too young to survive on their own in urban or rural areas of several municipalities of the Piedmont Region and died or were euthanized after admission to these facilities. The paired duodenal and liver samples of the 183 carcasses were frozen and transported to the Department of Veterinary Medicine, University of Teramo, and stored at −80 °C until virological investigation.

### 2.2. DNA Extraction from Tissue Samples

Duodenal and liver samples (1 g each) were homogenized using Mini Bead Mill (VWR International Srl, Milano, Italy) in 7 mL hard-tissue homogenizing mix tubes containing 4 mL of phosphate-buffered saline (0.15 M, pH 7.2). To further degrade the tissues and facilitate the release of virions into the solution, 0.5 mL of a 100 μg/mL proteinase K (1 mg/mL) was added to the samples and incubated at 37 °C for 3 h with shaking at 320 rpm. The reaction mixture was subsequently incubated at 60 °C for 15 min to inactivate the proteinase, and the supernatant, collected via centrifugation at 2500 rpm for 15 min, was stored at −20 °C. DNA was extracted individually from each duodenal (0.5 mL) and liver (0.5 mL) sample by using the TRIzol LS (Invitrogen, Ltd., Paisley, Scotland, UK), following the manufacturer’s instructions.

### 2.3. Detection of Parvoviruses

The presence of BoV DNA in hedgehogs’ duodenal and liver samples was assessed using a pan-BoV primer set, BoV-Fw 5′-GCCAGCACNGGNAARACMAA-3′ and BoV-Rev 5′-CATNAGNCAYTCYTCCCACCA-3′ [8], which enabled the amplification of a 141 bp fragment of the NS1 gene of members of the *Bocaparvovirus* genus (https://ictv.global/report/chapter/parvoviridae/parvoviridae/bocaparvovirus, accessed on 11 January 2024). Furthermore, a primer set (HhBoV_FW 5′-CCTGCCAGTACAGGMAAGAC-3′ and HhBoV_REV 5′-TTCATAAAATTWAGTTGAAC-3′), targeting a larger fragment of 385 bp at the NS1 gene, was designed based on the alignment of the sequences detected in this study and the corresponding conservative regions of the more genetically close related BoV sequences available on the GenBank database. The hedgehogs’ duodenal and liver samples were also screened by using a broadly reactive primer pair 165-Fw/371-Rev [7], which targeted a 207 bp region of the VP2 encoding gene of BuVs, members of the genus *Protoparvovirus* (https://ictv.global/report/chapter/parvoviridae/parvoviridae/protoparvovirus, accessed on 11 January 2024) (Table 1). For PCR amplification, the GoTaq^®^ Green Master Mix (Promega Italia S.r.l, Milan, Italy) and the suggested cycling thermal conditions were used.

### 2.4. Genome Sequencing and Phylogenetic Analysis

The amplified products were excised from the gel and purified using a QIAquick Gel Extraction Kit (Qiagen GmbH, Hilden, Germany). Sequencing was carried out bidirectionally with a 3730 DNA Analyzer using the BigDye Terminator Cycle (Applied Biosystems, Foster City, CA, USA). The basic Local Alignment Search Tool (BLAST; http://www.ncbi.nlm.nih.gov, accessed on 15 July 2023) and FASTA (http://www.ebi.ac.uk/fasta33, accessed on 15 July 2023) with default values were used to find homologous hits.

In order to collect further genetic data, attempts were made to obtain the complete genome of the two hedgehog BuV strains detected in our study. The PCR-amplified overlapping partial fragments of these strains were generated by using the primer sets previously described for protoparvovirus carnivoran 3 detection [7,11,21] and specific primers designed based on the available genome sequences (Table 1). PCRs were performed with TaKaRa La Taq polymerase (Takara Bio Europe S.A.S. Saint-Germain-en-Laye, France). The amplicons were purified and cloned using a TOPO XL Cloning Kit (Life Technologies, Carlsbad, CA, USA). Consensus sequences were generated by sequencing at least three clones for each PCR fragment and assembled via the sequential splicing of overlapping sequences. The alignment of the sequences was conducted using the MAFFT multiple alignment program version 7.388 plugin of the Geneious software version 11.0.18+10 (Biomatters Ltd., Auckland, New Zealand). Phylogenetic analyses were performed with the maximum likelihood method in the MEGA 11 software [22], using the Poisson model and supplying statistical support with bootstrapping of 1000 replicates.

## 3. Results

The molecular screenings for BoVs and BuVs revealed parvoviral DNA in 17 of the 183 hedgehog carcasses, with an overall prevalence of 9.3% (17/183). In detail, BoV DNA was identified in 15 animals (8.2%, 15/183), whereas BuV DNA was detected in 2 (1.1%, 2/183). Out of the 15 BoV-positive *E. europaeus* (3 from “La Ninna” and 12 from the C.A.N.C.), viral DNA was detected in duodenal samples from 10 animals (5.5%, 10/183), in the liver from 2 *E. europaeus* (1.1%, 2/183), and in paired intestinal and liver tissues from 3 animals (1.6%, 3/183). Using the pan-BoV strategy, we identified a total of 10 organs (2.7%, 10/366) as positive, whilst an additional series of 8 positive samples were identified upon rescreening with specific primers (4.9%, 18/366), with overall prevalence rates of 7.1% (13/183) and 2.7% (5/183) in the intestine and liver samples, respectively. Partial NS1 nucleotide sequences were obtained from PCR-positive tissue samples (Table 2). 

Through sequence comparison in the larger (385 nt) NS1 fragment, 18 amplicons showed 65.5–100% nt and 77.1–100% aa identities to each other. Specifically, a total of 14 BoV sequences (GenBank Accession No. OR682572-OR682585) detected in the duodenum and/or liver of 12 *E. europaeus* (592DU-2019, 617DU-2019, 618DU-2019, 1279DU-2019, 1079DU-2021, 1079L-2021, 1082L-2021, 1083DU-2021, 358DU-2022, 458L-2022, 637DU-2022, 637L-2022, 742DU-2022, and 1141DU-2022) had the highest similarity (98.3% to 99.4% nt and 99.1–100% aa) to two HhBoV strains, HeN-F4 and HeN-F1 (GenBank Accession No. OM451140-OM451141), recently reported in China in the *E. amurensis* [18]. Four BoV sequences (GenBank Accession No. OR682586-OR682589) detected in the duodenum of two animals (656DU-2022 and 657DU-2022) and in the liver and intestinal samples of an additional hedgehog (655DU-2022 and 655L-2022) showed the highest nt (98.9–99.4%) and aa (100%) identities to porcine bocavirus (PBoV) strains detected in pigs in China [23,24,25], South Korea [26], Belgium [27], East Africa [28], and the USA [29]. Furthermore, a high nt identity, ranging from 87.3% to 90.4%, was found for sequences related to PBoVs but detected in rats, minks, and mice [30,31,32,33,34]. Based on the inspection of the tree (Figure 1), the 14 HhBoV strains, identified in this study, were tightly clustered (bootstrap value, 100%) along with the recently identified Chinese HhBoVs [18], in a well-defined group. The four obtained PBoV-like sequences segregated with PBoVs detected in pigs genetically related to the strain PBoV H-18 (HQ291308), prototype of the species *Bocaparvovirus ungulate 4* (https://ictv.global/report/chapter/parvoviridae/parvoviridae/bocaparvovirus, accessed on 01 December 2023) [23,24,25,26,27,28,29], in a larger cluster including the bocaparvoviruses ungulate 4 related sequences detected in minks and rodents [30,31,32,33,34].

BuV DNA was detected only in duodenal samples of two *E. europaeus* (1.1%, 2/183), both recovered at the C.A.N.C. in 2019 and 2022, whilst the liver tissues tested negative. Through the sequence analyses of the 207 bp fragment of the VP2 coding region, the two strains showed 98.7% nt (98.6% aa) identity to each other and the highest identities (80.1–81.6% nt and 84.1–85.5% aa) to the HhBuV strains HeN-F1 and HeN-Ly (OM451142-43), previously detected in China in the Asian *E. amurensis* [18]. The nearly complete genome of the two Italian strains was obtained (GenBank Accession No. OR682590-OR682591). In detail, for the Hedgehog/331DU-2022/ITA strain, a genome sequence of 4664 bp was generated, including a partial 5′ untranslated region (UTR) (183 bp), the complete NS1 sequence (648 aa), the complete VP1 sequence (774 aa), and a partial 3′ UTR (223 nt). For the Hedgehog/1278DU-2019/ITA (4344 bp) strain, the 5′ partial NS1 gene (563 aa), the complete VP1 sequence (774 aa), and a partial 3′ UTR (212 nt) were detected. Based on genome sequence alignment, the two Italian HhBuVs displayed 99.6% nt identity to each other. The closest relative genome-wise (80.7–81.9% nt identities) was the HeN-F1 strain (OM451142) and, in the 5′ partial genome sequence, the HeN-Ly strain (OM451143) [18]. Nucleotide identities to the other members of the genus *Protoparvovirus* ranged from 39.3% to 59.2%. HhBuVs were more closely related to the bufaviruses identified in pigs (58.8% nt similarity) and dogs (57.5–58.2% nt) [7,35].

The genome coding sequence of the 331DU-2022/ITA strain, excluding the terminal UTR regions, was 4258 nt (GenBank Accession No. OR682590) with two major ORFs. The left ORF, coding for NS1, was 1947 nt, and the right ORF, encoding for VP1 and VP2, was 2325 nt. In the complete coding region, the Italian strain displayed an nt identity of 84.9% to the HhBuV complete sequence (strain HeN-F1) published in the databases [18], whilst identities to the other members of the genus *Protoparvovirus* ranged from 42.1% to 58.9%.

The NS1 of the Italian strain 331DU-2022 was characterized by a start codon MAV in an adequate Kozak sequence CACATG**G** [36]. Two conserved replication initiator motifs, namely ^131^GL**H**I**H**VLLQ^139^ and ^217^IYT**Y**FLQKN^225^ (conserved amino acids are in the boldface type), were identified [37]. In addition, highly conserved Walker motifs of the helicase domain, namely Walker A (^408^**GP**AST**GK**S^415^), B (^448^LIWI**EE**^453^), B′ (^465^**K**AIMS**G**QAIRLDQ**K**^478^), and C (^490^VIM**T**T**N**^495^), were detected [38,39]. The ORF1-encoded sequence showed 87.2% nt and 90.6% aa identities to the Chinese HhBuV, HeN-F1 strain, whilst identities to other parvovirus NS1 sequences were <63.8% nt and <63.5% aa. The termination of ORF1 overlapped the start of ORF2 by 14 nt. Through sequence alignment and comparison with other BuVs, two potential splice sites in the ORF1/ORF2 junction were found, a potential donor site (G¯GT) at nt 1961 (of coding sequence) and an acceptor site (AG¯G) at nt 2142. The putative VP1 sequence started at the end of ORF1 at nt 1934, upstream of the splice donor site at nt 1961. The phospholipase A2 (PLA2) motif was identified in VP1 N-termini with the expected highly conserved calcium-binding site (YLGPG) and catalytic residues (HD and D), detectable at amino acid positions 78–82, 101–102, and 123, respectively. The VP1 sequence showed 86.5–86.7% aa identity to the HhBuVs previously detected in China and <53.3% aa to the other species of the *Protoparvovirus* genus**.** The methionine codon of VP2 was located upstream of the glycine-rich sequence (GGGGGGGSGVG) that was also present in other parvoviral VP2 proteins.

Upon phylogenetic analysis based on the complete NS proteins (Figure 2a), the Italian and Chinese HhBuV strains (aa identity 90.6%) segregated into a well-defined group (bootstrap value 100%), which in turn fell into a defined cluster encompassing parvoviruses identified in rodents, dogs, sea otter, shrew, and pigs [7,14,21,35,40,41,42,43]. The aa identities within this group were 56.5–63.5%. The clustering (bootstrap value of 100%) of all the four HhBuVs (aa identity 86.5–99.6%) was confirmed in the VP1 capsid-based tree (Figure 2b), within a larger group including protoparvoviruses identified in dogs, sea otter, bats, pigs, and primates (overall aa identities 49.5–56.5%) [7,14,21,41,43,44,45,46,47,48,49,50].

## 4. Discussion

In the present study, we demonstrated that *E. europaeus* harbors several parvoviral species. The molecular screening of duodenal and liver samples, collected from a total of 183 carcasses of *E. europaeus* admitted to two Italian WRCs, revealed different parvoviruses belonging to either the genera *Bocaparvovirus* (8.2%, 15/183) or *Protoparvovirus* (1.1%, 2/183), with an overall prevalence of parvovirus DNA as high as 9.3% (17/183). BoVs were identified in both duodenal and liver samples. Indeed, ten *E. europaeus* had BoV DNA in the duodenum, two in the liver, and three in both tissue samples. Previously discovered viruses of the genus *Bocaparvovirus* have been frequently associated with respiratory, gastroenteric, and neurological diseases in animals and humans [16,51,52,53,54,55,56,57,58,59,60,61,62]. However, several studies have reported the detection of BoV DNA in the liver of animals, including dogs, marmots, and rats [6,34,63]. In addition, a link between BoVs and hepatitis has been strongly suggested in an immunodeficient child with clinical hepatitis but without respiratory symptoms and without other concomitant viruses [64].

Interestingly, the sequence analyses of the NS1 fragments obtained with both sets of primers revealed the circulation of two highly divergent BoVs (65.5–67.8% nt and 77.1–78.8% aa identities) in the investigated *E. europaeus* population. Indeed, the BoV sequences detected in 12 animals were genetically related (98.3% to 99.4% nt identities) to the HhBoVs identified in healthy Asian *E. amurensis* in China in 2020, during a viral metagenomic investigation in game animals [18]. Nonetheless, in the Chinese study, BoV sequence reads were generated from fecal samples [18]. Since the presence of the virus due to food contamination in the stools cannot be ruled out, this finding was not firmly demonstrated. Herein, however, the identification of HhBoV DNA in 10 duodenal, 2 liver, and 2 paired (intestine–liver) samples could indicate systemic spread (extraintestinal localization/replication), thus suggesting that *E. europaeus* might be a host species for this virus.

Three animals were infected with diverse BoV strains that showed the highest sequence match (98.9–99.4% nt identity) to the PBoVs detected in the stool or serum samples of pigs and classified in the species *Bocaparvovirus ungulate 4* [23,24,25,26,27,28,29]. Curiously, previous studies have already reported bocaparvovirus ungulate 4-related strains in other animals, including minks, shrews, house mice, and rats [30,31,32,33,34], and all these strains belong to the same viral species (aa identity in NS1 > 88%) [34]. The sequences detected in hedgehogs showed, in the short NS1 fragment, nt identities ranging from 87.3% to 90.4% to the PBoV-related strains. This could suggest the ability of the bocaparvovirus ungulate 4 to infect more animal hosts. Codon usage preference analysis based on the VP2 gene sequences of bocaparvovirus ungulate 4 strains detected in pigs and rodents indicated a higher adaptability of these viruses to rats rather than pigs, suggesting a possible origin from rodents [34]. Interestingly, in one of the three positive hedgehogs, parvoviral DNA was also identified in the liver sample. In rats, the bocaparvovirus ungulate 4 DNA has been detected in the liver and serum [34]. Also, previous studies found parvoviral DNA, simultaneously, in throat swabs, as well as fecal and serum samples from mice [33], and the lungs of rats [32], indicating a wide tissue tropism [34].

In our study, the lack of available blood samples prevented direct testing for viremia. Overall, BoVs have been mostly demonstrated as part of the enteric virome [16,51,52,53,54,55,56], but it cannot be excluded that the detected BoV strains’ replication may be occurring in hepatocytes or other liver cell types as well as in other extraenteric tracts. Furthermore, the pathogenetic potential of these viruses, if any, remains to be determined. The Chinese BoV-positive *E. amurensis* were healthy at the time of sampling, and, herein, for most animals, the cause of death was trauma, followed by predation, respiratory failure, and starvation. In addition, based on the macroscopic and histological findings of the available organs of parvovirus-positive animals, no significant changes were observed, with the exception of mild multifocal lymphoplasmacytic hepatitis in the liver of one HhBoV-positive *E. europaeus.* The screening for BuVs, members of the genus *Protoparvovirus,* revealed viral DNA in the duodenal samples of two *E. europaeus* (1.1%). The sequence analysis of the obtained short fragment (207 nt) of the ORF2 gene revealed the highest genetic identity (nt 80.1–81.6% and aa 84.1–85.5%) to the HhBuV strains detected in the Asian *E. amurensis* during a metagenomic investigation on game animals in China [18], indicating that these viruses are a common component of the hedgehog virome. The analysis of the complete genome sequence of the HhBuV 331DU-2022/ITA strain and the 5′ partial genome sequence of the 1278DU-2019/ITA strain confirmed the closest match (nt identity of 80.7–81.9% at the genome level) to the two Chinese HhBuV strains [18]. In the complete ORF1-encoded proteins, the Chinese HhBuV strain HeN-F1 (OM451142) and the Italian strain 331DU-2022/ITA (OR682590) displayed an aa identity of 90.6%, while identities to other members of the genus *Protoparvoviruses* were found to be lower than 63.5% aa. Based on the current criteria established by the International Committee on Taxonomy of Viruses, parvoviruses with >85% aa identity in the NS1 protein should be regarded as members of the same species. Accordingly, HhBuVs could be classified as a new species within the genus *Protoparvovirus,* for which we propose the name “*Protoparvovirus Eulipotyphla 2*”. Phylogenetic analyses based on NS and VP aa sequences confirmed the closest association between the Italian and Chinese HhBuVs. These viruses clustered within a larger group of protoparvoviruses commonly termed as BuVs and previously identified in other mammalian hosts, including pigs, shrews, rodents, fur seals, bats, non-human primates, and humans [7,14,21,35,40,41,42,43,44,45,46,47,48,49,50]. In the VP1 sequence, a difference of 13.3–13.5% aa was observed between the Chinese and Italian hedgehog BuVs, likely indicative of host species- or geographical-related patterns. For instance, genetic variability has been identified among human BuVs, classified into three distinct genotypes/serotypes [45,46,65,66], and among canine BuVs, classified into two genotypes [21,67].

Whether HhBuVs are associated with any disease in these animal species and the target organs/district remains to be established. In our study, the HhBuV-positive duodenal samples were collected from *E. europaeus* with an unknown death cause. Only one animal showed histological signs of moderate multifocal lymphoplasmacytic enteritis with the parasite infestation. Furthermore, the available paired liver samples were negative for HhBuV. In the Chinese investigation, one of the HhBuV strains detected was found in the fecal samples of a collection of 11 healthy *E. amurensis*, while the other positive sample was a lymph node of a deceased animal, but the clinical history was unknown. So far, BuV DNAs have been identified chiefly in the enteric tract of humans [65] and animals [21,35,40], but studies in non-human primates [44], shrews [42], sea otters [14], and domestic carnivores [7,11,41,50,68] reported the detection of viral DNA in respiratory samples and other tissues and organs, including serum, lymph nodes, spleen, lung and liver, suggesting the possibility of extraintestinal and/or systemic infections.

## 5. Conclusions

In conclusion, we gathered evidence on the circulation of parvoviruses of the genera *Bocaparvovirus* and *Protoparvovirus* in the European hedgehogs (*Erinaceus europaeus)*. Two genetically diverse BoV strains were identified in the surveyed *E. europaeus*, namely a major cluster related to *E. amurensis* BoVs and a minor cluster related to porcine BoVs. Also, we identified and reconstructed the nearly complete genome of the BuVs representing a novel protoparvovirus species. Our results warrant further studies to enhance the epidemiological data and assess the genetic diversity of these parvoviruses in *E. europaeus*. Furthermore, experimental attempts as well as a future screening of samples not tested as part of the current study such as brain, lung, spleen, and kidney tissues, including the in situ *hybridization* of tissues showing pathologies, will reveal whether both BoVs and HhBuVs can have a pathogenic role in the *E. europaeus* species’ intestinal or extraintestinal systems. Finally, the high mutation rates of BoVs and BuVs, as well as their recombination events, may lead to fast evolution and host escape, crossing interspecies barriers [69,70]. Considering the close contact of *E. europaeus* with humans and domestic animals, also providing a link to other wild species, gathering information on the virome of wildlife is pivotal not only for animal conservation but also for the assessment of zoonotic risks for humans from a One Health perspective.

## Figures and Tables

**Figure 1 microorganisms-12-00189-f001:**
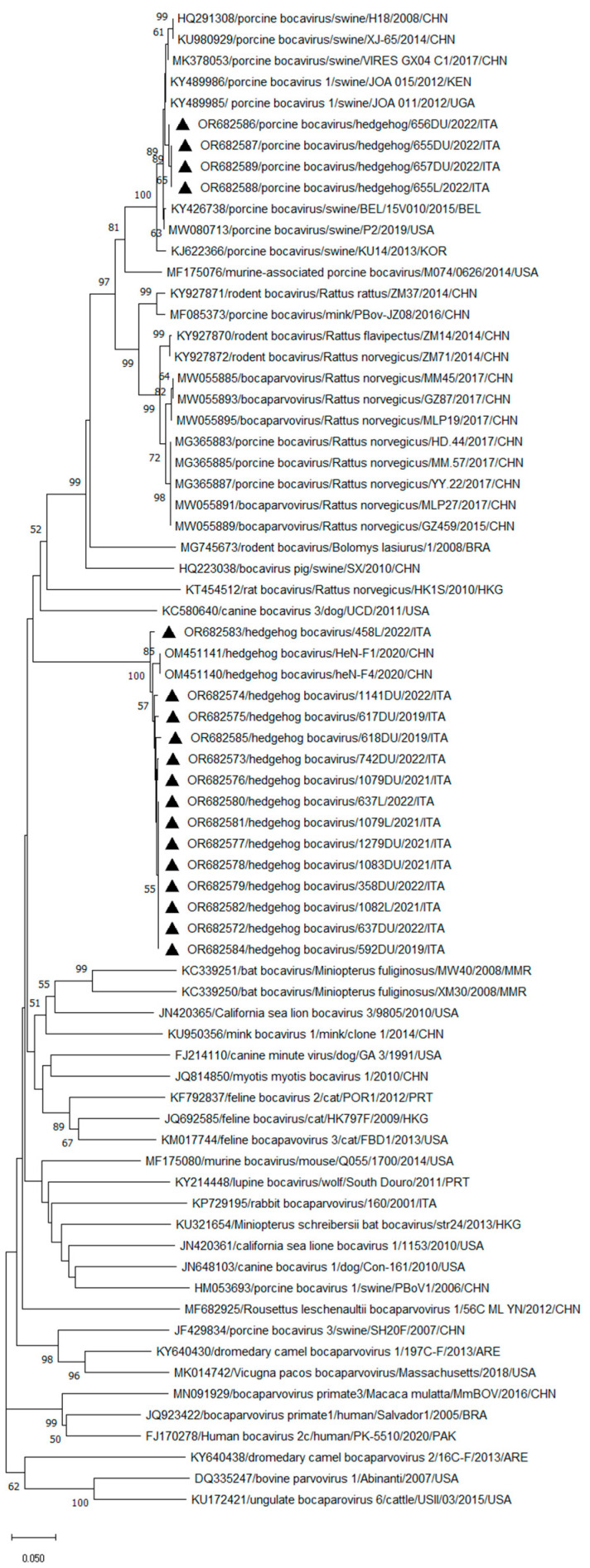
Phylogenetic tree based on the nt sequence of the partial NS1 gene of the BoV strains detected in this study. A selection of representative sequences of the genus *Bocaparvovirus* was retrieved from GenBank. The tree was generated using the neighbor-joining method and Kimura 2-parameter model supplying statistical support with bootstrapping of 1000 replicates. Bootstrap values > 50% are shown. Black triangles indicate the BoVs identified in this survey.

**Figure 2 microorganisms-12-00189-f002:**
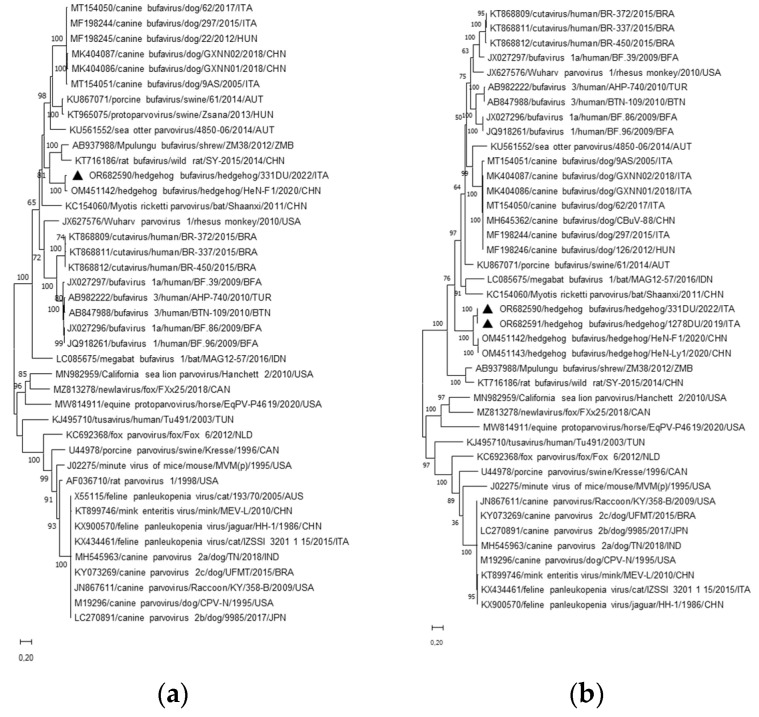
Phylogenetic analyses based on the aa sequence of the complete NS1 (**a**) and VP (**b**) of the HhBuVs identified in this study. The trees, constructed with a selection of protoparvovirus strains representative of each species, were generated using the maximum likelihood method based on the Poisson correction and supplying statistical support with bootstrapping of 1000 replicates. The scale bar indicates nucleotide substitutions per site. Black triangles indicate the HhBuV strains detected in this study.

**Table 1 microorganisms-12-00189-t001:** List of primers used in this study for detection and characterization of BuVs. Nucleotide position refers to the sequence of the HhBuV strain HeN-F1 (GenBank Accession No. OM451142).

Oligonucleotide	Position	Sequence (5′-3′)	Sense	Use	References
CPPV 165F	3068–3087	CTGGTTTAATCCAGCAGACT	+	Screening PCR	[7]
CPPV 371R	3257–3274	TGAAGACCAAGGTAGTAGG	−	Screening PCR	[7]
HhBuV 5′F	1–23	GGAGGGGGCCTCTTACGTCATCA	+	Sequencing	This study
HhBuV 141F	141–163	ATGAAGAAACCTACAAATATAGC	+	Sequencing	This study
CPPV 453R	653–673	CCCCATTTTTCTGCAAAGWAT	−	Sequencing	[21]
CPPV 1142F	1352–1374	TACCATGCAATCATSTGCTGCCT	+	Sequencing	[21]
CPPV 1397R	1607–1632	TTGCCTTTTTGATCAAGTCTGATTGC	−	Sequencing	[21]
CPPV 1409F	2622–2643	TCATATTCCTGGAGAAACATCA	+	Sequencing	[11]
HhBuV 2670R	2649–2670	AGGGCTTACCTCTTTTGGTGCC	−	Sequencing	This study
CPPV-L3-F	3134–3160	TGAACAAGAAATAGACAACATTGTCAT	+	Sequencing	[7]
CPPV-L3-R	3208–3231	AAAGAGCAGTTAGGTCATTGTTGT	−	Sequencing	[7]
CPPV 1414R	3550–3571	ACTGGCAATCTAACAGACATAT	−	Sequencing	[11]
HhBuV 3854F	3854–3873	TGACTACAACCACGGAGACC	+	Sequencing	This study
HhBuV 4000R	3981–4000	TTCCAGCTGGTTGTGTGTGC	−	Sequencing	This study
CPPV 1571R	4421–4439	TTATAGAGTAATATTAGGC	−	Sequencing	[11]
HhBuV 4631R	4612–4631	GGTGTCAAAGTGACTTTGAA	−	Sequencing	This study

**Table 2 microorganisms-12-00189-t002:** BoV strains detected in *E. europaeus* in this study.

Virus	Strain	Tissue	GenBank Accession Number
Hedgehog bocavirus	637DU-2022	duodenum	OR682572
	742DU-2022	duodenum	OR682573
	1141DU-2022	duodenum	OR682574
	617DU-2019	duodenum	OR682575
	1079DU-2021	duodenum	OR682576
	1279DU-2019	duodenum	OR682577
	1083DU-2021	duodenum	OR682578
	358DU-2022	duodenum	OR682579
	592DU-2019	duodenum	OR682580
	618DU-2019	duodenum	OR682581
	637L-2022	liver	OR682582
	1079L-2021	liver	OR682583
	1082L-2021	liver	OR682584
	458L-2022	liver	OR682585
Porcine bocavirus	656DU-2022	duodenum	OR682586
	655DU-2022	duodenum	OR682587
	655L-2022	liver	OR682588
	657DU-2022	duodenum	OR682589

## Data Availability

The data supporting the findings of this study are openly available in the GenBank database.

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
