# Peer review of "Molecular Surveillance for Bocaparvoviruses and Bufaviruses in the European Hedgehog (Erinaceus europaeus)"

_microorganisms, 2024, doi:10.3390/microorganisms12010189_

Round 1

Reviewer 1 Report

Comments and Suggestions for Authors

The manuscript deal with the molecular identification of some parvoviruses species in the European hedgehogs inhabiting Piedmont Region (Italy). It should be noted that the sample of hosts used in the study was quite large. The data obtained by the authors is especially important due to the ability of bocaparvoviruses to cause respiratory, gastroenteric and neurological diseases in humans and animals.Congratulations to the authors on their efforts to carry out high-quality scientific work.

The manuscript undoubtedly makes a certain contribution to our knowledge of the distribution of bocaparvoviruses and bufaviruses in nature, and undoubtedly meets the goals and objectives of the journal Microorganisms and can be published.

However, I have some remarks about this manuscript.

In the section Results the Genbank accession numbers should be presented in the form of a table. This way they will be more indicative.

Usually, species or genera of living organisms are specified in italics (according International Code of Zoological Nomenclature (ICZN). Family names do not need to be italicized. Besides, at the first mention of species in text of article, its full Latin name with the author and year of description should be given. For example: Erinaceus europaeus Linnaeus, 1758, etc.

Please remember to use article “the” when mentioning of animal species (the European hedgehog, the Amur hedgehog).

According MDPI rules, in the titles of manuscript, sections and subsections, words are written with a capital letter (Molecular Surveillance for Bocaparvoviruses and Bufaviruses in the European Hedgehog (Erinaceus europaeus); . 2. Materials and Methods; 2.2. DNA Extraction from Tissue Samples  …..).

 Perhaps the name of the subfamily of viruses (Protoparvovirinae) should be included in the title of the publication?

In the section Materials and Methods it is necessary to indicate what viruses classification the authors used.

Line 58 – better write “… in the Amur hedgehog Erinaceus amurensis Schrenk, 1858 …”

Line 63 – better use “… in the European hedgehog Erinaceus europaeus Linnaeus, 1758…”

Further in the text, try to use only the Latin names of animals (hedgehogs). In scientific articles, try to avoid using the usual names of animals, or use them only at the first mention along with the Latin names.

When the Latin name of an animal is mentioned for the second time in the text, the generic name should be abbreviated.

Line 76 – italics E. europaeus

Line 329 - Erinaceus europaeus.

Author contributions (line 337) must be made in accordance with the journal's guidelines. Authors' initials should be used here.

I express my opinion – the manuscript can be published, but minor corrections are needed.

Author Response

We thank the Reviewer 1 for his/her comments/suggestions.

Reviewer 1 (R1)

The manuscript deal with the molecular identification of some parvoviruses species in the European hedgehogs inhabiting Piedmont Region (Italy). It should be noted that the sample of hosts used in the study was quite large. The data obtained by the authors is especially important due to the ability of bocaparvoviruses to cause respiratory, gastroenteric and neurological diseases in humans and animals. Congratulations to the authors on their efforts to carry out high-quality scientific work.

The manuscript undoubtedly makes a certain contribution to our knowledge of the distribution of bocaparvoviruses and bufaviruses in nature, and undoubtedly meets the goals and objectives of the journal Microorganisms and can be published.

However, I have some remarks about this manuscript.

 R1.1 - In the section Results the Genbank accession numbers should be presented in the form of a table. This way they will be more indicative.

Reply to R1.1 – Many thanks to the Referee 1 for this suggestion. A table (Table 2) was added in the section “Results” of the revised manuscript.   

R1.2 - Usually, species or genera of living organisms are specified in italics (according International Code of Zoological Nomenclature (ICZN).Family names do not need to be italicized. Besides, at the first mention of species in text of article, its full Latin name with the author and year of description should be given. For example: Erinaceus europaeus Linnaeus, 1758, etc.

Reply to R1.2 – We thank Referee 1 for providing us with advice on how to correctly apply the International Code of Zoological Nomenclature in our publication. We truly appreciate it and will treasure it.

R1.3 - Please remember to use article “the” when mentioning of animal species (the European hedgehog, the Amur hedgehog).

Reply to R1.3 – Many thanks to the Referee 1. The article “the” was used in the revised manuscript when the animal species were mentioned.

R1.4 - According MDPI rules, in the titles of manuscript, sections and subsections, words are written with a capital letter (Molecular Surveillance for Bocaparvoviruses and Bufaviruses in the European Hedgehog (Erinaceus europaeus); . 2. Materials and Methods; 2.2. DNA Extraction from Tissue Samples…..).

Reply to R1.4 – Many thanks to the Referee 1 for this observation. All words in the titles of manuscript, sections and subsections were written with a capital letter in the revised manuscript.

R1.5 - Perhaps the name of the subfamily of viruses (Protoparvovirinae) should be included in the title of the publication?

Reply to R1.5 – We thank the Referee 1 for this suggestion. However, in order to be concise as Authors instructions required and to avoid an excessive use of brackets, if possible, we would not want to include the subfamily in the title.

R1.6 - In the section Materials and Methods it is necessary to indicate what viruses classification the authors used.

Reply to R1.6 – In the revised manuscript, in the “Material and Methods” section, “Detection of Parvoviruses” subsections, the links to the web pages of the International Committee on Taxonomy of Viruses (ICTV) of the genera Bocaparvovirus and Protoparvovirus were added.

R1.7 - Line 58 – better write “… in the Amur hedgehog Erinaceus amurensis Schrenk, 1858 …”

Reply to R1.7 – Many thanks to the Referee 1. We followed this suggestion in the revised manuscript.

 R1.8 - Line 63 – better use “… in the European hedgehog Erinaceus europaeus Linnaeus, 1758…”

Reply to R1.8 – Many thanks to the Referee 1. We also followed this suggestion in the revised manuscript.

R1.9 - Further in the text, try to use only the Latin names of animals (hedgehogs). In scientific articles, try to avoid using the usual names of animals, or use them only at the first mention along with the Latin names.

Reply to R1.9 – We thank the Referee 1 for this advice that we followed throughout the revised manuscript.

R1.10 - When the Latin name of an animal is mentioned for the second time in the text, the generic name should be abbreviated.

Reply to R1.10 – We thank again the Referee 1 for this advice that we followed throughout the revised manuscript.

 R1.11 - Line 76 – italics E. europaeus

Reply to R1.11 – As suggested by the Referee 1, “…European hedgehogs (Erinaceus europeaus)” was replaced by “E. europaeus”, in the revised manuscript.

R1.12 - Line 329 - Erinaceus europaeus.

Reply to R1.12 – In the revised manuscript “Erinaceus Europaeus” was corrected with “Erinaceus europaeus”.

 R1.13 - Author contributions (line 337) must be made in accordance with the journal's guidelines. Authors' initials should be used here.

Reply to R1.13 – As suggested by the Referee 1, Authors’ initials were used in Author contributions the in the revised manuscript in accordance with journal's guidelines.

Reviewer 2 Report

Comments and Suggestions for Authors

Dear Authors,

The article presents a detailed phylogenetic analysis and sequencing of the genome of some viruses in the studied animals. The study carried out is presented in a manner supported by a recent, varied bibliography, which addresses the subject of this study.

Parvoviruses are a highly pathogenic species, sometimes causing severe and often fatal epizootics of gastroenteritis worldwide, which is remarkably supported in this article by the parallel made between the animal species studied Erinaceus europaeus and other species. However, in the discussion section, lines 262-263, the systemic spread of the genes of this species of viruses is supported in the studied hedgehogs. The authors should support this statement and present other evidence to strengthen the statement that based on only 2 samples (liver) and 10 duodenal, European hedgehogs are the host species of the virus. Possibly expand the evidence, so that this statement can be supported, especially since later (lines 284 285) the cause of death of these animals is presented as trauma, starvation, etc. The identification of HhBuV viruses as a new species among Protoparvoviruses is important, that's why I support the above, for lines 262-270. In the conclusion section, I suggest the authors to additionally argue the importance of this HhBuV genomic sequencing study in order to have a clearer picture related to the risk assessment on the human species.

Author Response

We thank the Reviewer 2 for his/her comments/suggestions.

Reviewer 2 (R2)

The article presents a detailed phylogenetic analysis and sequencing of the genome of some viruses in the studied animals. The study carried out is presented in a manner supported by a recent, varied bibliography, which addresses the subject of this study.

 R2.1 - Parvoviruses are a highly pathogenic species, sometimes causing severe and often fatal epizootics of gastroenteritis worldwide, which is remarkably supported in this article by the parallel made between the animal species studied Erinaceus europaeus and other species. However, in the discussion section, lines 262-263, the systemic spread of the genes of this species of viruses is supported in the studied hedgehogs. The authors should support this statement and present other evidence to strengthen the statement that based on only 2 samples (liver) and 10 duodenal, European hedgehogs are the host species of the virus. Possibly expand the evidence, so that this statement can be supported, especially since later (lines 284 285) the cause of death of these animals is presented as trauma, starvation, etc.

Reply to R2.1 – We are in agree with the Referee 2. Unfortunately, only these paired tissue samples were available for each hedgehog carcasses included in this study. We are planning a new investigation, that will include the collection of additional samples as brain, lung, spleen and kidney in order to map more in detail the tissue tropism of BoVs. Based on the suggestions of the Referee 2, we modified the discussion section in the revised manuscript, as follows. The sentence “In our study, however, HhBov DNA was detected in 10 duodenal, 2 liver and 2 paired (intestine-liver) samples, indicating systemic spread (extra-intestinal localization/replication) and suggesting that European hedgehogs are a host species of this virus.” was replaced with “Herein, however, the identification of HhBoV DNA in 10 duodenal, 2 liver and 2 paired (intestine-liver) samples, could indicate systemic spread (extra-intestinal localization/replication) hence suggesting that the E. europaeus might be considered as host species for this virus.. Furthermore, the sentences “BoVs have been mostly demonstrated as part of the enteric virome [16, 51-56], but it cannot be excluded that replication occurs in other extra-enteric sites. In our study, for most animals the cause of death was a trauma, followed by predation, respiratory failure, and starvation. Based on the macroscopic and histological findings of the available organs of the parovirus-positive animals, no significant changes were noticed, with exception of mild multifocal lymphoplasmacytic hepatitis in the liver of one HhBoV-positive hedgehog.” were replaced as follows: “In our study, the lack of available blood sample prevented direct testing for viremia. Overall, BoVs have been mostly demonstrated as part of the enteric virome [16, 51-56], but it cannot be excluded that detected BoV strains replication may be occurring in hepatocytes or other liver cell types as well as in other extra-enteric tracts. Furthermore, a pathogenetic potential of these viruses, if any, remains to be determined. The Chinese positive hedgehogs were healthy at the time of sampling and, herein, for most animals the cause of death was trauma, followed by predation, respiratory failure, and starvation. In addition, based on the macroscopic and histological findings of the available organs of the parovirus-positive animals, no significant changes were noticed, with exception of mild multifocal lymphoplasmacytic hepatitis in the liver of one HhBoV-positive E. europaeus.”. Finally, the sentence Our results warrant further studies to increase the epidemiological data and to assess the genetic diversity of these parvoviruses in the E. europaeus. Furthermore, experimental attempts as well as future screening of samples not tested as part of the current study as brain, lung, spleen, and kidney, including in situ hybridization of tissues showing pathologies, will reveal whether both BoVs and HhBuVs can have a pathogenic role in the E. europaeus intestinal or extraintestinal systems.was added in the “Conclusions” section of the revised manuscript.

R2.2 - The identification of HhBuV viruses as a new species among Protoparvoviruses is important, that's why I support the above, for lines 262-270.

Reply to R2.2 – As above, we are in agree with the Referee 2. Accordingly, we added, in the “Conclusions” section of the revised manuscript, the sentence “Our results warrant further studies to increase the epidemiological data and to assess the genetic diversity of these parvoviruses in the E. europaeus. Furthermore, experimental attempts as well as future screening of samples not tested as part of the current study as brain, lung, spleen, and kidney, including in situ hybridization of tissues showing pathologies, will reveal whether both BoVs and HhBuVs can have a pathogenic role in the E. europaeus intestinal or extraintestinal systems.” (see also Reply to R2.1).

 R2.3 - In the conclusion section, I suggest the authors to additionally argue the importance of this HhBuV genomic sequencing study in order to have a clearer picture related to the risk assessment on the human species.

Reply to R2.3 – In the “Conclusions” section we considered that the zoonotic transmission potential deserves attention for both BoVs and BuVs. Indeed, the high mutation rate and ability to recombination of these viruses may facilitate fast evolution and host escape. Also, considering the strict social interactions between humans, pets, wildlife and hedgehogs it will be important to collect data on the prevalence of various pathogens, including zoonotic ones in hedgehogs. Accordingly, the sentence “Gathering information on the virome of wildlife animals is pivotal for animal conservation and also, in the One Health perspective, to assess the zoonotic risks for humans.” was modified as follows Finally, BoVs and BuVs high mutation rate as well as recombination events may lead fast evolution and host escape, crossing interspecies barriers [69, 70]. Considering the close contact of E. europeaus with humans and domestic animals, also providing a link to other wild species, gathering information on the virome of wildlife is pivotal for animal conservation and also, in the One Health perspective, to assess the zoonotic risks for humans.” in the revised manuscript.